# Gully rehabilitation in Southern Ethiopia – value and impacts for farmers.

Wolde Mekuria[1], Euan Phimister[2,5], Getahun Yakob[3], Desalegn Tegegne[1], Awdenegest Moges[4], Yitna Tesfaye[4], Dagmawi Melaku[1], Charlene Gerber[5], Paul D. Hallett[6], Jo U. Smith[6]

1 International Water Management Institute (IWMI), Nile Basin and East Africa Office, P. O. Box 5689, Addis Ababa, Ethiopia.

2 Business School, University of Aberdeen, Aberdeen AB24 3FX, UK.

3 Southern Agricultural Research Institute, P.O. Box 6, Hawassa, Southern Nation Nationality People Regional State, Ethiopia

4 Hawassa University, P. O. Box 5, Hawassa, Ethiopia.

5 Stellenbosch Business School, Stellenbosch University, Cape Town, South Africa.

6 School of Biological Sciences, University of Aberdeen, Aberdeen AB24 3UU, UK.

*Correspondence to*: Wolde Mekuria (w.bori@cgiar.org)

**Abstract.** Gully erosion can be combatted in severely affected regions like sub-Saharan Africa using various low-cost interventions that are accessible to affected farmers. For successful implementation, however, biophysical evidence of intervention effectiveness needs to be validated against the interest and priorities of local communities. Working with farmers in a watershed in Southern Ethiopia, we investigated (a) the effectiveness of low-cost gully rehabilitation measures to reduce soil loss and upward expansion of gully heads, (b) how farmers and communities view gully interventions, and (c) whether involving farmers in on-farm field trials to demonstrate gully interventions improves uptake, knowledge and perceptions of their capacity to act. On-farm field experiments, key informant interviews, focus group discussions and household surveys were used to collect and analyse data. Three gully treatments were explored, all with riprap, one also with grass planting, and one with grass planting and check-dam integration. Over a period of 26 months these low-cost practices ceased measurable gully head expansion, whereas untreated gullies had a mean upward expansion of 671 cm resulting in a calculated soil loss of 11.0 tonnes. Farmers had a positive view of all gully rehabilitation measures explored. Ongoing rehabilitation activities and on-farm trials influenced knowledge and understanding of similar gully treatments among survey respondents. On-farm experiments and field day demonstrations empowered farmers to act, addressing pessimism from some respondents about their capacity to do so.

**1 Introduction**

Land degradation caused by gully erosion is one of the most persistent and complex global environmental problems (Poesen
et al. 2003; Menendez-Duarte et al. 2007), resulting in a significant loss of key ecosystem services, such as the provision of
food and water and regulation of climate (SDG Report 2019). The global extent of gully erosion includes severely degraded
land in the Loess Plateau of China (Cheng et al. 2007), the northwestern highlands of Ethiopia (Dagnew et al. 2017) and
selected areas of the United States (Bernard et al. 2010), where it accounts for up to 70% of soil loss. Multiple factors contribute
to the initiation and development of gullies and gully erosion in a landscape; both environmental (e.g. drought and concentrated
runoff) (Tebebu et al. 2010; Conforti et al. 2011; Conoscenti et al. 2013), and anthropogenic factors (e.g. land use and land
cover change, overgrazing, deforestation, and repeated cultivation) (Moges and Holden, 2009; Jahantigh and Pessarakli 2011;
Asmamaw et al. 2012; Asres et al. 2016; Alem 2022).

Although gully erosion is a global problem that occurs in all geographical areas (Menendez-Duarte et al. 2007; Ionita et al.
2015), Africa is the worst-hit continent (Were et al. 2023). This could partly be attributed to declining and low vegetation
cover that has led to increased overland flow over less stable soils (Ireri et al. 2021). Around 20 to 25% of the land area of
Sub-Saharan Africa (SSA) is severely affected by gully erosion (FAO 2001). Ground measurements of gully erosion in Tigray
in northern Ethiopia found that it accounted for 28% of the average total soil loss of 14.8 t ha$^{-1}$ y$^{-1}$ measured over a 4-year
period (Nyessen et al. 2008). In southern Ethiopia, ground measurements of soil erosion within gullies ranged from 11 to 30 t
ha$^{-1}$ yr$^{-1}$ (Belayneh et al. 2024). The economic costs of soil loss due to gully erosion are high (Ayele et al. 2015); for Kenya,
this was estimated to be equal to total agricultural exports (equivalent to $ 390 million annually or 3.8% of GDP) (Cohen et
al. 2006). For the Ethiopian highlands, the total cost of gully erosion, estimated over two years, was over $18, 000, or $17 per
person per year or about 19% of per capita income (Ayele et al. 2015).

These costs are experienced mostly by rural communities, particularly poorer households, who generate much of their income
from the land (Mekuria et al. 2023). Severe gully erosion affects the livelihoods of rural communities in several ways, including
the degradation of croplands (Yitbarek et al. 2012; Yazie et al.2021), land fragmentation (Frankl et al. 2011), reduced livestock
carrying capacity of rangelands (Adimas et al. 2021; Alem 2022), limitation of movement (Worku and Tripathi 2015) and
potentially death of both humans and livestock (Moges and Holden, 2008). Off-site gully erosion increases siltation of
freshwater ecosystems, which can degrade and destroy water infrastructure, such as hydroelectric dams (Degife et al. 2021).

Clearly controlling gully erosion is a major environmental challenge, but this can be difficult to achieve since, once the soil
has degraded and gullies have formed, complex measures are required to stop their expansion. The rehabilitation of large
gullies usually takes several decades and requires huge financial resources (Poesen et al. 2003; Nasri et al. 2008; Perroy et al.
2010). In line with this, a study in Ethiopia (Water and Land Resource Center, 2015) indicated that, on average, the total costs
of gully rehabilitation and development using both structural and vegetative measures amounts to US$ 17,180 per hectare.
However, the costs could vary greatly depending on the availability of the construction materials, site and gully characteristics,

and the type of interventions. This, in turn, suggest that available technologies need to be contextualized to a specific site to be effective, considering costs, labour and available resources in addition to the physical challenge (Blake et al. 2018; Shi et al. 2019; Wen et al. 2021; Bai et al. 2023). In poorer countries the challenge of controlling gully erosion is greater due partly to the lack of context-specific technologies, material and financial resources and technical skills (Yitbarek et al. 2012; Erkossa et al. 2015).

Some of these challenges can be averted if gully erosion mitigation occurs at its onset and focuses on gully head expansion. Across Ethiopia, for instance, such gully erosion controlling measures have been implemented as part of the national soil and water conservation initiatives since the 1980s (Haregeweyne et al. 2015). These initiatives include Food-for-Work (FFW) (1973–2002), Managing Environmental Resources to Enable Transition to more Sustainable Livelihoods (MERET, 2003– 2015), Productive Safety Net Programs (PSNP, 2005–present), Community Mass Mobilization through free-labor days (1998– 70 present), the National Sustainable Land Management Project (SLMP, 2008–2018) and Climate Action Through Landscape Management (2019 – present). However, despite decades of interventions, gully erosion has worsened rather than improved, with more communities affected than ever before (Steenmans 2017; Belayneh et al. 2020). In northwestern Ethiopia, for example, the temporal changes of 44 gullies (Belayneh et al. 2020) measured from a time series of high-resolution satellite images and field surveys found increased gully length (from 3.83 to 10.04 km), density ($0.71–1.87$ km km$^{-2}$), surface area 75 damage (3.31–11.42 ha), and soil loss ($60.6$ to $273.1 \times 10^3$ t) over a period of 2001–2018. Such increases are partly attributed to limited gully rehabilitation practices. Context specific measures (e.g. measures that consider availability of resources and capacities of local communities) are clearly needed to implement gully rehabilitation (Kropacek et al. 2016). To be effective and widespread, farmers need to be involved. Rabinovich et al. (2022) demonstrated that providing better knowledge and understanding can increase farmers' intention to adopt measures to prevent soil erosion.

Previous soil and water conservation interventions show that farmers and communities are more likely to adopt and maintain measures when scientific and local knowledge and circumstances are well integrated (Amsalu and de Graaff 2006; Haregeweyn et al. 2015). Ethiopian farmers are aware of the importance of gullies on soil erosion. However, addressing their impacts has typically arisen from participation in communal land development activities, instead of measures a farmer might take on their own land (Menginstu and Assefa 2020). For individual farmers, soil and water conservation measures can be perceived as 85 requiring too much labour to be implemented on their own land (Bewket 2007).

To counter this, numerous field experiments in Ethiopia have explored whether low-cost interventions, accessible to individuals or small groups of farmers on their own land, are effective in preventing and rehabilitating gullies at an early stage of their development. Barvels and Fensholt (2021) found that, in the Highlands of Ethiopia, vegetative measures effectively rehabilitated gullies. Further research in this area by Belay and Bewket (2012) and Addisie et al. (2018) indicated that 90 vegetative measures combined with physical structures could effectively control gully erosion and convert gullies to productive land. Physical measures on their own, such as re-grading gully heads and banks, and adding stone riprap at the gully heads have been demonstrated to stop the upward expansion of gully heads and control the expansion of gullies (Addisie et al. 2018).

Farmers therefore have accessible physical and biological options to mitigate gully erosion, with positive outcomes to the environment likely to be providing personal benefit too. In research to date, however, there is a disconnect between farmers' perceptions and their willingness to adopt effective gully erosion mitigation strategies. The most effective strategies need to draw together socioeconomic drivers with biophysical evidence, such as effectiveness of interventions to halt upward expansion of gully heads and reduce soil loss. To address this gap, the key contribution of this study arises from the combination of such on-farm field experiments with a range of quantitative and qualitative analyses to explore how farmers and communities view gully interventions and whether demonstrating on farm field experiments in-context changes their knowledge and perceived capacity to act.

Although a range of practices are deployed globally to tackle gully erosion, in the global south, low-cost options that can be established by communities are far more common. One area where these interventions are being tested is Halaba district in southern Ethiopia, which is particularly prone to gully erosion, with most of the agricultural landscapes containing small to large gullies (Mekuria et al. 2023). We use this region as a case study to interpret both biophysical and socioeconomic drivers of gully rehabilitation measures. First, the effectiveness of low-cost gully rehabilitation measures to reduce the upward expansion of gully heads and soil loss are explored using a series of on-farm field experiments. Drawing on these findings, the understanding of the community and individuals of gully management practices was then explored. This was by key informant interviews to provide base information to assess the drivers, pressures, state, impact and responses for gully erosion in the communities, while focus group discussions were used to assess the benefits and costs of multiple gully rehabilitation measures from the perspective of the local communities. Finally, household survey information from similar areas with and without the on-farm field experiments was used to test the hypothesis that effectively demonstrating low-cost control measures, combined with dissemination activities within the community, would change knowledge, perceived capacity, and intention to take measures to address gullies. By including farmers in our investigation of gully mitigation strategies, our goal was to identify policies and practices that would be the most effective, acceptable and practical to implement.

## 2 Methods

### 2.1 Study area

This study was conducted in Aba-Bora watershed, Halaba, Southern Ethiopia (Figure 1). Selected characteristics of the watershed are summarized in Table 1 (Mekuria et al. 2023). Households in the Aba-Bora watershed make their livelihood mainly from a subsistence mixed crop-livestock farming system (Mekuria et al. 2023). A number of households also engage in off-farm and non-farming activities, but agriculture and crop sale income accounts for over 80% of household incomes (Mekuria et al. 2023).

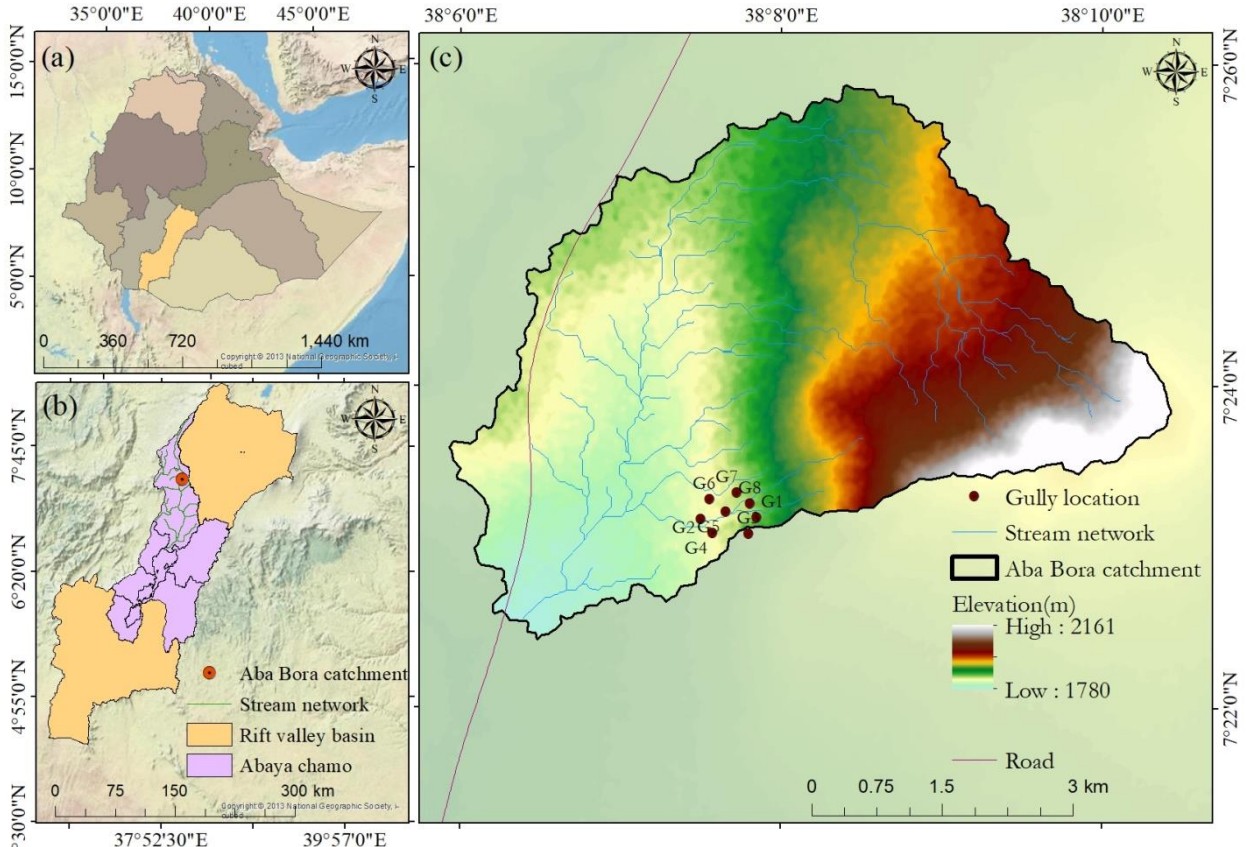

**Figure 1:** Location map of the study area, Aba-Bora watershed, Halaba, Southern Ethiopia: (a) rift valley lakes basin within Ethiopia, (b) Abaya Chamo subbasin within rift valley lakes basin, (c) Aba Bora watershed. *Source: Authors' creation*

Declining soil fertility, severe soil erosion, reduced access to surface and groundwater, and poor water quality are the main socio-economic and environmental challenges in the studied watershed (Sinore and Umer 2021). In particular, gully erosion (Figure 2) is one of the main causes of land degradation, manifested by deep gullies exacerbated by increased deforestation in the upstream areas (Mekuria et al. 2023), soils susceptible to soil erosion (Yakob et al. 2022) and a shift from livestock to crop-based agriculture (Byg et al. 2017). The lack of low-cost gully rehabilitation technologies and the awareness of local communities for addressing gullies at the early stage of gully initiation and formation have also contributed to the expansion of gullies (Addisie et al. 2017).

**Table 1**. Selected characteristics of Aba-Bora watershed (Mekuria et al. 2023).

| Site and household characteristics | Values |
|---|---|
| Area (km$^2$) | 28.91 |
| Elevation (m) | From 1780 to 2161 |
| Dominant land use and land cover | Croplands |
| Rainfall distribution | Bimodal |
| Duration of long rainy season | July – September |
| Duration of short rainy season | March - May |
| Monthly rainfall during long rainy season (mm) | Ranges from 100 to 146 |
| Monthly rainfall during short rainy season (mm) | Ranges from 20 to 143 |
| Annual rainfall (mm) | Ranges from 752 to 1272 |
| Mean annual temperature (ºC) | Ranges from 19 to 22 |
| Average Education (Years in School) | 2.182 |
| Average Family size | 6.472 |
| Average landholding (Ha) | 0.961 |
| Tropical Livestock Units | 2.985 |

**2.2 Gully erosion study design and data collection**

Nine paired gullies (i.e. treated and untreated) were selected (Table 2) from the midslope position (Figure 1). They all had depths less than 3 m to control the influence of gully size and location on the low-cost gully rehabilitation measures investigated. Moreover, paired plots had similar soil, topography and land use and land cover, with spatial distribution of less than two hectares of land area so that climatic variables, such as rainfall and temperature were similar. The study considered three low-cost gully rehabilitation measures as treatments (Figure 2), each replicated three times. The treatments were (1)

regrading the gully head (45º) by establishing stone riprap (Riprap treatment); (2) regrading the gully head and gully bank by establishing stone riprap at the gully head and planting grasses on the gully banks (Riprap-grass planting integration treatment); and (3) regrading the gully head and gully bank by establishing stone riprap at the gully head, planting grasses on the gully banks and constructing small sandbag check-dams (Riprap-grass planting-check-dam integration treatment). The on-farm field experiment was established in 2021. Low-cost gully rehabilitation measures, in this study, refers to interventions that can be

implemented with farmers capacity or with limited external support.

In the established field experiments, the upward expansion of gully heads in the treated and control gullies were monitored at different intervals, after 4, 9, 12, 17 and 26 months. The upward expansion of gully heads was measured against pegs installed at 1 and 2 meters from the gully head. The distance of a permanent reference point (i.e. big trees on the farm and close to the investigated gullies) from the 2 m peg was also measured in case it was removed by the upward expansion of the gully head with time. To estimate the volume and amount of soil loss during the period of investigation, the depth, width and length of

150 the upward expanded part of gullies was measured at the beginning and end of the project period at 26 months. The soil loss from the upward expansion of gullies was calculated by multiplying the volume of soil by an approximate bulk density of 1.3 g cm$^{-3}$. Erosion further down the gully was beyond the scope of our study.

**Table 2**. Dimensions of treated and control gullies prior to the establishment of on-farm trials.

|  | Replication one | | Replication two | | Replication three | |
|---|---|---|---|---|---|---|
|  | Treated | Control | Treated | Control | Treated | Control |
| **Treatment one** | | | | | | |
| Average depth (m) | 2.7 | 2.7 | 1.2 | 1.2 | 1.3 | 0.8 |
| Average width (m) | 5.8 | 5.3 | 2.8 | 3.8 | 2.5 | 2.1 |
| Average length (m) | 27.0 | 23.0 | 16.0 | 18.0 | 25.0 | 27.0 |
| Coordinates | | | | | | |
| X (Longitude) (degree) | 38.134535 | 38.184920 | 38.132695 | 38.133720 | 38.132406 | 38.132295 |
| Y (Latitude) (degree) | 7.383361 | 7.323133 | 7.381327 | 7.383768 | 7.3839 | 7.384105 |
| Elevation (meter) | 1856 | 1868 | 1853 | 1866 | 1856 | 1860 |
| **Treatment two** | | | | | | |
| Average depth (m) | 2.2 | 0.8 | 2.1 | 1.5 | 1.1 | 0.7 |
| Average width (m) | 4.3 | 1.8 | 5.3 | 5.0 | 2.1 | 1.2 |
| Average length (m) | 26.5 | 34.0 | 20.2 | 17.0 | 22.0 | 14.0 |
| Coordinates | | | | | | |
| X (Longitude) (degree) | 38.132318 | 38.132303 | 38.133043 | 38.132303 | 38.132540 | 38.132470 |
| Y (Latitude) (degree) | 7.382736 | 7.382705 | 7.382736 | 7.382736 | 7.384143 | 7.384105 |
| Elevation (meter) | 1847 | 1845 | 1864 | 1845 | 1850 | 1862 |
| **Treatment three** | | | | | | |
| Average depth (m) | 1.7 | 1.9 | 1.3 | 0.8 | 2.8 | 3.5 |
| Average width (m) | 4.6 | 4.5 | 3.4 | 3.3 | 4.6 | 4.5 |
| Average length (m) | 24.6 | 26.0 | 32.6 | 28.0 | 25.4 | 15.0 |
| Coordinates | | | | | | |
| X (Longitude) (degree) | 38.13575 | 38.13575 | | | 38.133720 | 38.133658 |
| Y (Latitude) (degree) | 7.38 | 7.38 | | | 7.383768 | 7.383731 |
| Elevation (meter) | 1875 | 1876 | | | 1866 | 1854 |

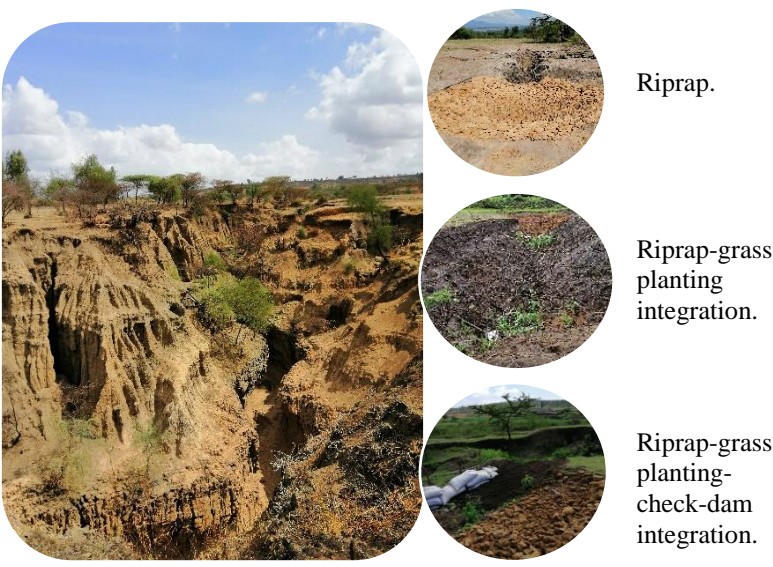

Riprap.

Riprap-grass planting integration.

Riprap-grass planting-check-dam integration.

**Figure 2.** Severe gully erosion in Aba-Bora and low-cost mitigation options *Picture credit @Wolde Mekuria*

According to Yakob et al. (2024, unpublished), the soil in the study area are Luvisols, with an argic subsoil horizon containing more clay than in the topsoil, and a high base saturation at 50–100 cm depth. Selected characteristics of the soil in the study area are summarized in Table 3.

**Table 3**. Physicochemical properties of soils in the study area.

| Horizon | Depth (cm) | %Sand | %Clay | %Silt | Textural class | pH | CEC Meq / 100 g soil | %OC | P mg/kg | Na mg/kg |
|---------|-----------|-------|-------|-------|----------------|------|---------------------|------|---------|----------|
| A | 0-26 | 38 | 28 | 34 | Clay loam | 6.82 | 43.52 | 4.44 | 4.26 | 141.18 |
| Bt1 | 26-43 | 14 | 44 | 42 | Silty clay | 7.18 | 43.74 | 3.83 | 3.95 | 316.63 |
| E | 43-110 | 46 | 24 | 30 | Loam | 7.75 | 42.22 | 2.59 | 3.01 | 558.31 |
| Bt2 | 110-132 | 18 | 44 | 38 | Clay | 7.38 | 51.63 | 2.14 | 5.52 | 580.18 |
| E' | 132-180 | 18 | 28 | 54 | Silty clay loam | 7.70 | 51.48 | 2.01 | 3.55 | 538.46 |
| Bt3 | 180-200+ | 14 | 36 | 50 | Silty clay loam | 7.37 | 49.51 | 1.85 | 4.45 | 473.30 |

### 2.3 Stakeholder engagement and qualitative data collection and analysis

Four different forms of stakeholder engagement and qualitative data collection were conducted: (i) field days; (ii) key informant interviews involving experts; (iii) focus group discussions; and (iv) household surveys. Details of these activities are summarised below, with details provided in the supplementary information.

*Field days:* To promote knowledge and understanding of the experiments, a field day was organized in the kebeles where the on-farm trials were established. This was held in November 2022 in collaboration with the Southern Agricultural Research Institute (SARI) and district and zonal agricultural offices. Kebeles are typically localities containing around 500 households, with an administrative centre around which agricultural extension is typically organized. The field day focused on the demonstration of gully head treatments and gully bank stabilization, as well as incentives used to support the sustainable management of land resources. More than 55 participants, comprising farmers, extension agents, experts from district, zonal and regional levels agricultural offices, and local administrative bodies, attended the field day to share ideas.

*Key informant interviews:* Key informant interviews were conducted to understand the system and assess the drivers, pressures, state, impact and response (DPSIR) to gully erosion and develop the conceptual DPSIR framework of the interventions for gully erosion and formation. The DPSIR approach was adopted as it has been widely deployed to provide a causal framework to link environmental and social processes (Kristensen 2004). Twenty-five key informants consisting of 8 local practitioners (district and zonal agricultural offices), 10 knowledgeable farmers, 2 NGOs, 2 cooperatives and 2 civil societies/community-based organizations and 1 local administrative body were selected. We used semi-structured questionnaires and checklists to gather data. The discussion points with key informants included causes and severity of gully erosion, existing gully rehabilitation measures, effectiveness of responses to address gully erosion, management and allocation of natural resources, and the key changes following the implementation of gully rehabilitation measures. Knowledgeable farmers in this study were experienced in organizing and leading watershed management practices, members of community watershed team and early adopters.

*Focus group discussions:* Two focus group discussions (separate men and women) assessed the costs and benefits of five gully rehabilitation measures from the perspective of local communities. The assessed gully rehabilitation measures included both those in our field experiments and others commonly implemented locally; i.e. gully-head treatment, gully reshaping and planting grasses, loose rock check-dams, vegetation log check-dams and gabion check-dams. The focus groups first discussed the selected gully rehabilitation measures (i.e. the practices were presented using understandable descriptions in the local language and confirmed by the participants), specifically the advantages and disadvantages from the perspective of the local communities. Next, the participants rated the importance of each identified advantage and disadvantage, giving scores of 3 points for the most important, 2 for the intermediate and 1 for the least important one. Finally, the participants graded each practice or intervention, as positive, negative, or neutral.

The qualitative data collected through key informant interviews and focus group discussions were analysed using content analysis (Hsieh and Sannon 2005; Bernard 2006; Elo and Kyngas 2008). For qualitative analyses, we developed a coding scheme based on the main thematic areas, including items such as drivers of gully erosion and formation, pressure on the natural resources, response of gully erosion, impacts of gully rehabilitation measures and decision-making processes. These were coded manually. Deductive manual coding, which starts with a predetermined set of codes derived from the conceptual framework used in this study (i.e. the DPSIR framework) was then employed using codes such as severity of gully erosion, types of drivers and impacts, and types of interventions. During the coding process, more codes were added when new issues emerged from the textual data. In addition, codes were merged, removed, or modified to avoid repetition and resolve disagreements between codes.

*Household surveys:* In addition, two household surveys were conducted in 2021 and 2023 in the Aba-Bora and nearby Guder watersheds in localities where land was judged to be highly degraded with large gullies present. This included the kebele where the on-farm field experiment and focus group discussion took place. In 2021, data were collected from 522 households (including 123 from the kebele where the on-farm field experiment took place). In 2023, a follow-up survey collected data from 500 households, of which 451 could be identified as the same respondents (including 114 where the on-farm field experiment took place). The sample represented a random sample of households drawn from lists provided by each kebele administration after stratification based on wealth and gender status. The data were collected during February and March in each of 2021 and 2023 using a team of enumerators employing computer tablets, with the questionnaires available in both English and the local language Amharic (Supplementary material 1).

The 2021 and 2023 household surveys included a series of questions exploring respondents' knowledge, attitudes and behaviour with respect to land degradation and gullies. The respondents' data compared "treated" and "non-treated" to explore the potential impact of the on -farm experiment. Where identical questions were asked in both 2021 and 2023 surveys, the impact of the on-farm experiment was examined using a panel difference in difference (DD) approach, controlling for

individual fixed effects and a common trend (Glewwe and Todd, 2022, see also supplementary materials 1 and 2). The DD model compares changes in the indicator over time. Therefore, the DD model compares trends in the control group from before and after the project to trends in the treatment group. The double difference then refers to the difference over time (the first difference) and difference between the control and treatment (the second difference). If the trends are statistically significantly greater for the treatment group (in a statistical sense), this suggests that the intervention had an impact. Thus, the DD estimator combines cross-sectional and over-time variation to correct for differences between groups when treated and controls start from different level. Supplementary material 1 and 2 provides further details of the questions asked and the DD method.

The 2023 survey also included several more specific questions (items) on gullies, drawing on the nature of the on-farm experiments, to explore exploring respondents' knowledge, understanding, perceived capacity to act, and behaviour with respect to land degradation and gullies. Knowledge, understanding and perceived capacity to act can be conceived as latent (unobservable) values, but where different survey questions (scales or items) can be designed to capture these underlying unobserved values (see supplementary materials 1 and 2).

The specific approach applied here draws on the marketing literature to consider both recall and recognition of different interventions (Boshoff and Gerber 2008; Macdonald and Sharp 2000; Hoyer and Brown 1990). This approach has been also used to explore how knowledge and understanding may relate to attitudes about nature conservation (Pearson et al. 2022; Veríssimo et al. 2017; Schlegel and Rupf 2010). In line with this approach, the respondents were first asked in an open-ended question format to name three interventions or activities that they thought were important in reducing land degradation. The answers were then grouped and weighted to provide an awareness score ranging from 0 for completely unaware to 3 for very aware (Boshoff and Gerber 2008). The second set of questions explored the extent to which the respondents have knowledge of specific interventions. Here the respondents were shown pictures of different interventions, where the first two (gully head treatment and reshaping gully banks etc) linked directly to treatments one and two implemented in the on-farm experiments, while the third and fourth interventions shown could be more loosely related to treatment three (Table 2). First, to explore respondents' recognition and understanding, they were asked their view of the effectiveness of the measures. Second, to explore their perception of their capacity to implement the measures, they were asked whether they could undertake these measures on their own, or if they needed the help of neighbours or full community mobilization (see supplementary materials 1 and 2 for the specific question formats used). The original 5-point answers were aggregated into the three categories reported on (on own, with neighbours, community). The reliability to which the various sets of questions (items) captured the same underlying latent variable (knowledge, understanding and capacity to act) was tested using Cronbach's alpha. Cronbach's alpha is an internal consistency method of reliability, that shows a measure's homogeneity and occurs when different attempts of measuring a concept converge on the same result (Churchill, 1979). The coefficient alpha ranges from 0 to 1, and coefficient alpha of 0.7 or greater is considered to be an acceptable measure of reliability (Taber, 2018).

The more specific questions on gullies asked in 2023 only were analysed using a simple comparison between the treated and non-treated kebeles. This included analysing the respondents' capacity to implement the measures by estimating a simple

multinomial logit to model the probability that a respondent felt interventions could be implemented on their own, needed neighbours, or needed community mobilization (Supplementary material 2).

## 3 Results

### 3.1 Gully head upward expansion and soil loss

Regardless of the mitigation practice, all treated gullies had negligible upward expansion (Figure 3). This contrasted with
255 untreated gullies that expanded by a few centimetres after 4 months, increasing up to several metres after 26 months, ranging from 19 (± 4.3) to 671 (± 354) cm. Assuming a bulk density of 1.2 – 1.5 g cm$^{-3}$, the estimated soil loss due to the upward expansion of gully heads in the untreated gullies ranged from 2.8 to 20.2 tonnes, with a mean value of 11.0 (± 5.9) tonnes in 26 months. This value was zero in the treated gullies.

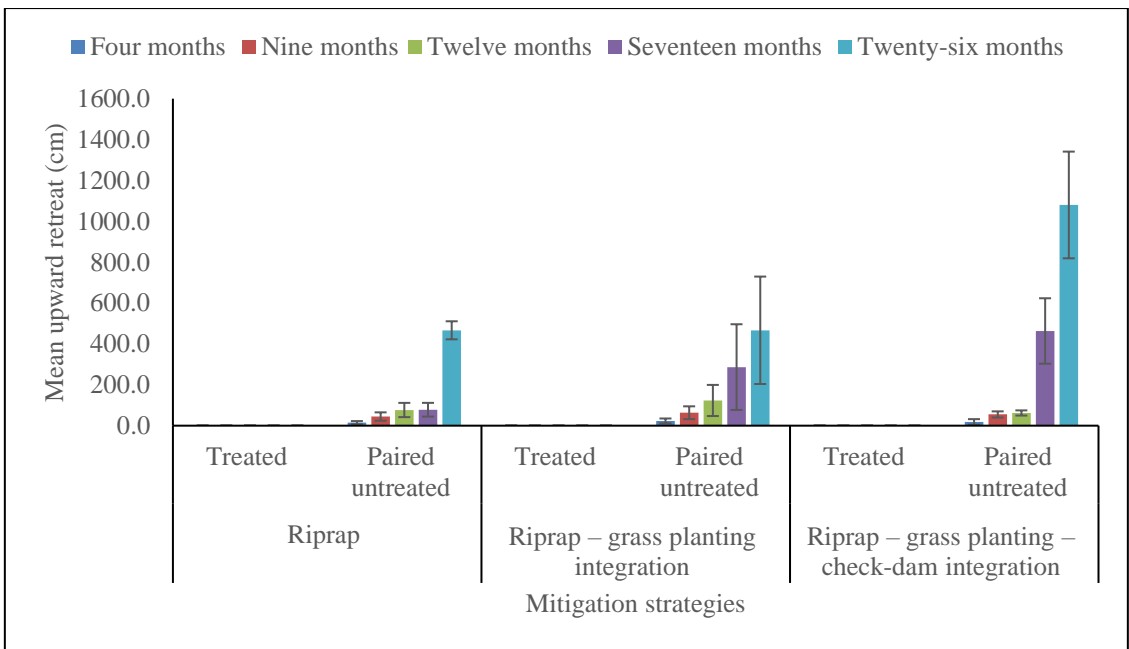

**Figure 3:** The mean upward expansion of gully heads in the treated and paired untreated gullies as described in the study design. The paired untreated gullies accounted for their spatial distribution, where a grouped control sites would introduce variability due to slight changes in climate or soils. For all treated gullies, the values for the upward expansion of gully heads was negligible.

### 3.2 The drivers, pressures, state, impact and response for gully erosion and formation

Good knowledge of gully initiation processes was demonstrated by key informants. They indicated that gullies usually initiated
in the upstream areas and took the form of rill erosion. Over time they recognised that rills got longer and deeper in the mid- and foot-slope landscape positions to form gullies. They remarked that the density and depth of gullies usually increased from mid-slope to foot-slope positions, but that gullies were unevenly distributed within the watershed with a few locations eroding differently along the slope. The key informants identified multiple human-induced, natural and climatic drivers of gully erosion

and formation (Figure 4). About 80% of the key informants characterized the severity of gully erosion as very severe, while the rest characterized gully erosion as severe, causing environmental, social and economic impacts. Gully erosion was considered to cause both long- and short-term negative impacts that affect the livelihood of local communities (Figure 4). The key informants also mentioned multiple initiatives implemented in the watershed to address gully erosion. These initiatives covered multiple dimensions; capacity building, providing technical support and implementing diverse interventions or measures (Figure 4). The commonly implemented gully rehabilitation measures included check-dams (loose rock, sandbags and gabion check-dams), biological measures such as planting multipurpose tree species and grasses on gully banks, and gully head treatments.

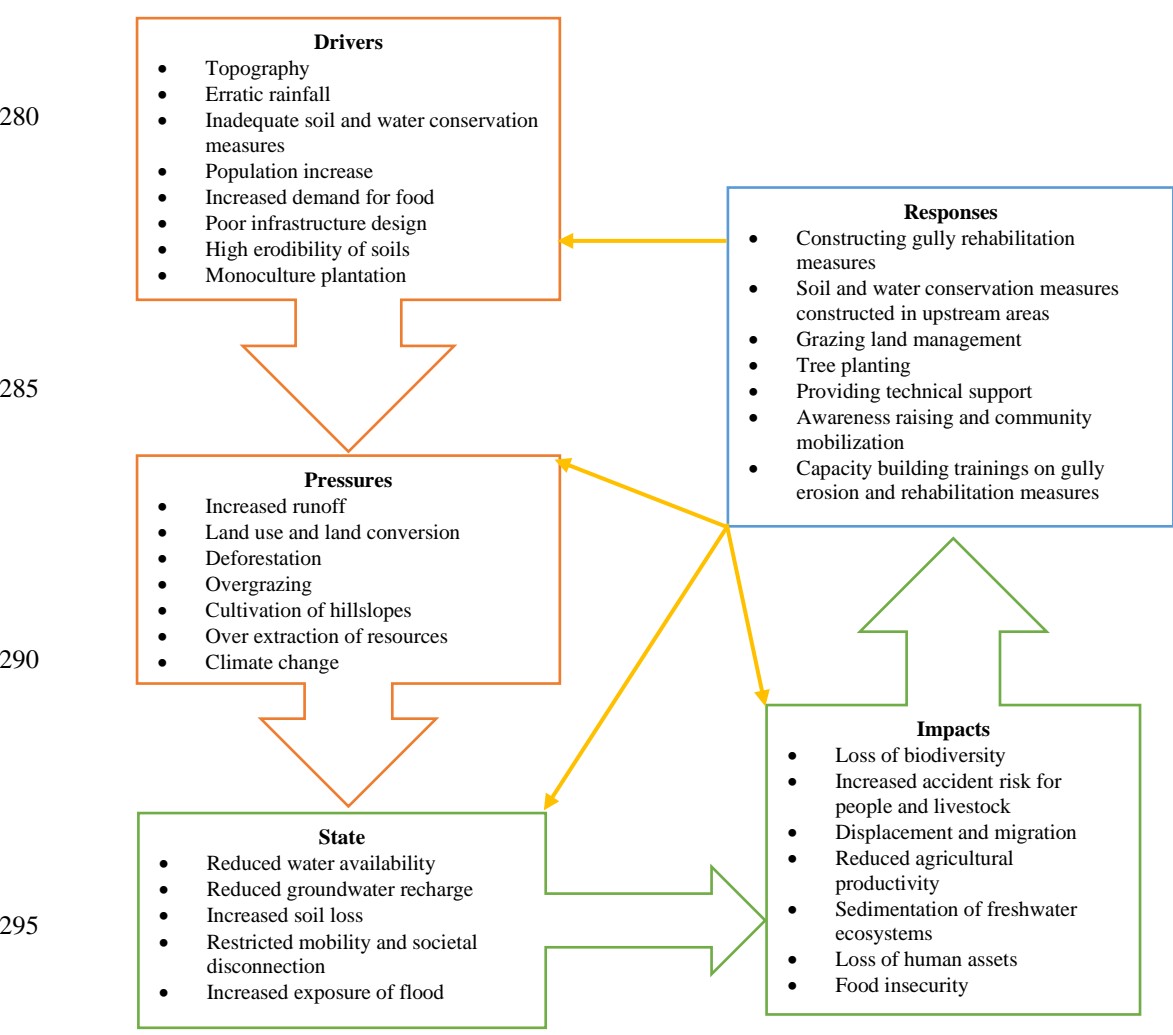

**Figure 4:** Conceptual drivers, pressures, state, impact and response framework for gully erosion and formation taken from key informant interviews. The yellow arrows in the figure show that a response, which is the result of an undesired impact can affect any part of the chain between driving forces and impacts.

The majority (90%) of key informants indicated that the implemented gully rehabilitation activities were successful, but that they did not cover all affected areas and were inadequate compared with the severity of the problem. They further elaborated that the rehabilitations measures, not only reduced soil erosion, but could also convert degraded lands to productive lands. The restoration of degraded lands by gully rehabilitation included reduced expansion and upward expansion of gullies, reduced runoff and soil loss, and improved cultivated and grazing land management (Figure 5). Furthermore, key informants suggested that gully rehabilitation measures enabled farmers to diversify their crops to grow cereals, vegetables and fruit. This was mainly attributed to reduced soil loss that resulted in enhanced soil nutrient retention and sometimes the accumulation of sediments behind check-dams. One of the key informants elaborated this as:

*"Farmers who migrated have started to come back and grow cash crops such as khat, vegetables and maize due to the improvement of soil fertility. Particularly, this is observed at the downstream areas of each treated gullies".*

In relation to the contribution of gully rehabilitation measures to livestock production, one of the key informants stated:

*"Decades ago, households used to have more than 30 livestock on average, however, the expansion of overgrazing and shortage of livestock feed forced them to reduce the number of livestock. Following the recent community-based watershed development activities, including gully rehabilitation works, the number of livestock per household is increasing due to increased production and availability of livestock feed from restored areas and gullies".*

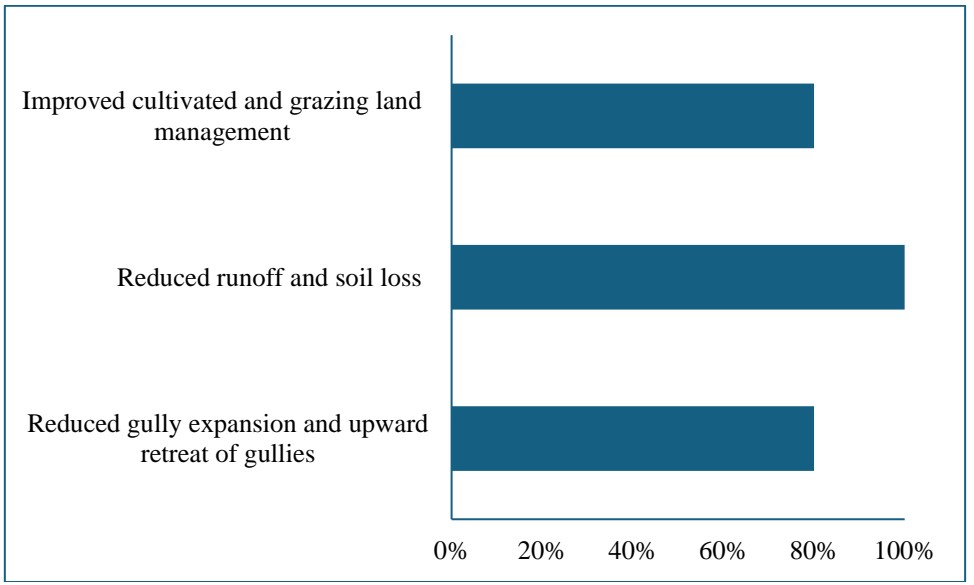

**Figure 5:** Mechanisms identified by key informants by which gully rehabilitation measures restore degraded lands. Numbers indicate the proportion of key informants that mentioned the specific mechanisms. The sum of numbers is greater than 100% as a respondent can mention multiple ways.

The ongoing rehabilitation activities were also key to establishing community awareness about the negative consequences of gully erosion and the need for collective action to control gully erosion. The key informants mentioned that effective

implementation and sustainability of gully rehabilitation measures required continuous technical support, particularly if more challenging biological measures were adopted. However, a few (10%) of the key informants stressed that it was hard to judge the success of interventions compared to the challenges of gully erosion. The majority (90%) of key informants also indicated that the efforts were not adequate compared with the severity of the problem.

The key informant interview results suggested that gully erosion can be addressed by individual or collective actions. Particularly, the respondents mentioned that small gullies (of less than 3 m depth), such as those assessed in our field research, can be rehabilitated using locally available materials and capacities. However, once the gullies get large and deep, it is beyond the capacity of local communities to rehabilitate them. The decisions on the planning, design and implementation of gully rehabilitation measures combined both bottom-up and top-down approaches. At the local or kebele level, a community watershed team was the key decision-maker, dictating interventions areas, types of interventions, and the labor and material contributions of communities. However, there were also cases where the regional bureau and zonal departments allocated a quota (e.g. types and quantity of interventions) to each district and kebele. The regional and zonal departments and NGOs were also involved in providing technical, material and financial support.

Community level decisions in relation to gully rehabilitation measures usually depend on the contribution of free labor and locally available materials. Our survey found this extended beyond building materials, such as stone and wood, to equipment and tools required for construction. In line with this, the community watershed team worked closely with kebele and district level experts to decide on community mobilization. This determined who participated, the types of interventions and the use of material and labor. There were also household-level decisions that could be implemented by individual households, such as taking measures to prevent runoff from entering their houses and farmlands. On and around their farmlands these usually involved constructing wooden or vegetation check-dams, water harvesting structures and bunds. Some individuals identified by the key informants were aware of gully erosion and took their own actions around the homestead by planting trees or by gully head treatment.

Cooperative efforts had also been implemented to effectively manage gully rehabilitation measures, including informal platforms established by Idir (self-support groups) and Iqub (voluntary cooperatives). Some actions were instigated by bylaws, so worked closely with formal institutions to enforce laws and protect interventions from free-riders (i.e. those who seek benefits while violating existing rules and regulations. The cooperative efforts had incentive mechanisms, such as engaging landless young people and women in income generating activities through providing beehives, sheep and oxen. Such incentive mechanisms strengthened the protection of land management practices including gully rehabilitation measures in a way that could be sustained and beneficial to farmers. Furthermore, the use of locally available materials, and the involvement of elders and religious institutions in awareness creation campaigns, were effective strategies to sustainably implement gully rehabilitation measures.

The key informants mentioned multiple sources of information. A majority (40 - 60%) of the respondents indicated that the key sources of information were the ongoing gully rehabilitation activities, personal experience and observation. The main limitation of gully rehabilitation measures constructed using locally available materials was the short lifespan of physical interventions, such as sandbags or log check-dams. However, when physical interventions were combined with vegetation, the longevity of the intervention improved. Our findings suggest that most of the gully rehabilitation measures were well received

by the communities, with the only concern being a temporary loss of access to their land or resources during construction. Out-scaling or wider implementation of gully rehabilitation measures was constrained by a shortage of personal tools (mentioned by 30% of the respondents), limited access to inputs and raw materials (30%), increasing prices of inputs (e.g. gabion wires, 60%), high labor demand (10%), technical requirement (30%) and limited sources of finance (10%).

### 3.3 Costs and benefits of gully rehabilitation measures from the perspective of local communities

Participants of focus group discussions, representing men and women separately, identified 24 benefits (or advantages) and 17 costs (or disadvantages) of the five gully rehabilitation measures assessed in the Aba-Bora watershed (Supplementary materials 3 and 4). Across both focus groups, the number of benefits identified for most of the gully rehabilitation measures outweighed the costs (Figure 6). There were gender differences in perceived costs and benefits. While female groups identified equal numbers of costs and benefits for loose rock check-dams, male groups felt the benefits were greater than the costs. An opposite

trend between genders was observed for gabion check-dam (Figure 6), where men indicated higher costs than women groups. The groups of both women and men identified more benefits for gully reshaping and planting, indicating that there was a consensus among the two groups regarding the benefits of this intervention. Both groups identified comparable numbers of benefits and costs.

    The benefits were diverse (Supplementary material 3) and covered environmental (e.g. conservation and increased access to

water, improved soil fertility, and reduced runoff and soil erosion), economic (e.g. increased agricultural production, reduced farmland loss and increased availability of livestock feed) and social (e.g. movement of people and livestock, reduced damage by flood, and knowledge exchange) categories. There were 24 identified benefit factors, but women were the only respondents to identify (i) the use of locally available materials, (ii) conserving and increasing access to water, and (iii) increased access to productive land. On the other hand, the men's groups listed 8 benefits that were not identified by women's groups. These were

(i) increased income, (ii) increased access to diverse food, (iii) improved appearance of the landscape, (iv) reduction in damage caused by floods, (v) regulation of micro-climate, (vi) ease of constructing the interventions, (vii) durability of the interventions, and (viii) serving as a learning site (Supplementary material 3). The participants also identified several costs of the assessed interventions or practices that were environmental (e.g. lack of effectiveness in reducing soil erosion and runoff), economic (e.g. high labor demand, cost, shortage of local available inputs) or social (e.g. demand in skilled manpower,

technical support) (Supplementary material 4).

Across both groups, the number of environmental benefits identified by the participants were larger than the respective environmental costs (Figure 7). Whereas female groups identified more social costs than benefits, male groups had an opposite opinion (Figure 7). Across both groups, the benefits of the assessed gully rehabilitation measures mainly related to the categories of environmental and economic benefits, whereas the costs largely related to the categories of economic and social costs (Figure 7).

The local surveys suggested that the benefits outweighed the costs for most of the interventions (Figure 6). The ratings of the importance of benefits assigned by women's groups across all the interventions varied between 2.3 and 2.5 (i.e. the rating is out of 3, and 3 represents high importance, 2 – medium and 1-low importance), with a mean value of 2.4, while the ratings of the costs ranged from 1.8 to 2.5, with a mean value of 2.0. The ratings of benefits assigned by men's groups varied between 2.2 and 2.8, with a mean value of 2.5, while the ratings of the costs ranged from 2.4 to 2.9, with a mean value of 2.6.

Apart from gabion check-dams, all other gully rehabilitation measures assessed were evaluated as positive (Table 4). The local communities considered environmental and economic factors more than social factors in their overall assessment of the gully rehabilitation measures (Table 4).

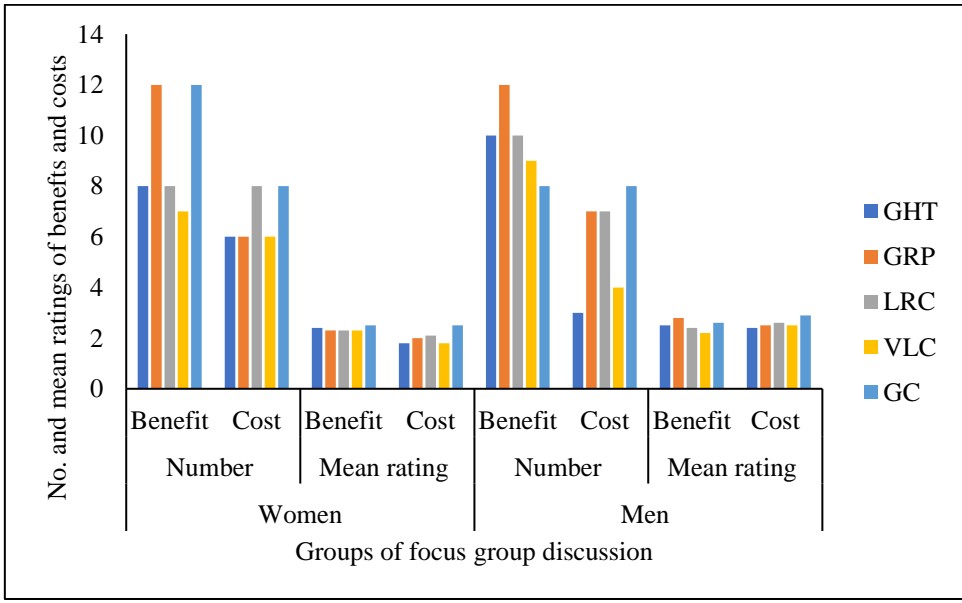

**Figure 6:** The number and ratings of the importance of benefit and cost factors in the perspective of the local communities. GHT – gully head treatment, GRP – gully reshaping and planting, LRC – loose rock check-dam, VLC – vegetation log check-dam, GC – Gabion check-dam. The values indicate the aggregate numbers of benefits and costs of all the three categories (environmental, economic, and social).

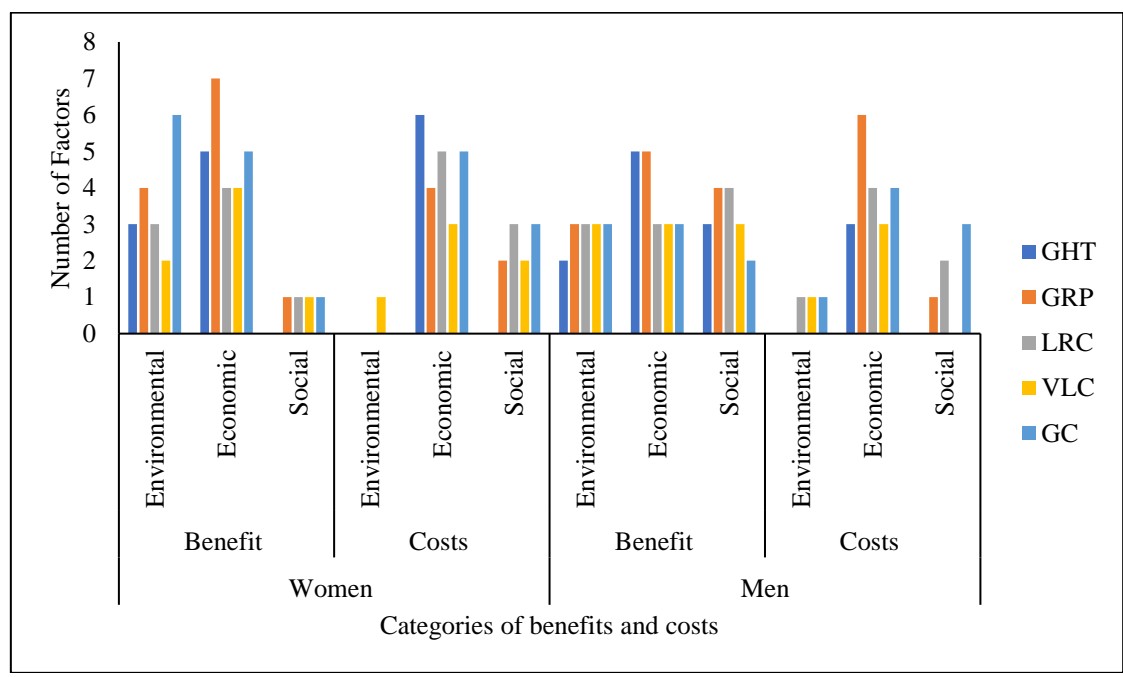

**Figure 7:** The number of benefit and cost factors under each of the categories of benefits and costs. For abbreviations, refer to Figure 6.

**Table 4**. Local communities' overall assessment of gully rehabilitation measures.

| Gully rehabilitation measures | Overall assessment | Reasons |
|---|---|---|
| **Women** | | |
| Gully head treatment | Positive | Good water retention and soil conservation. |
| Gully reshaping and planting | Positive | Good for increased access to grass for livestock. |
| Loose rock check-dam | Positive | Good performance and has the advantage of water collection. |
| Vegetation log check-dam | Positive | Easy to construct. |
| Gabion check-dam | Negative | The high cost of materials and the technical requirement, which is beyond the capacity of the local communities. |
| **Men** | | |
| Gully head treatment | Positive | Practical and relatively easy to construct. |
| Gully reshaping and planting | Positive | Additional benefit from grass (for livestock). |
| Loose rock check-dam | Positive | Good performance and longer lifespan compared to VLC. |
| Vegetation log check-dam | Positive | Easiest to construct and with locally available materials. |
| Gabion check-dam | Negative | The high cost of input materials and its technical requirement, which is beyond the capacity of local communities. |

### 3.4 Local communities' knowledge, attitudes and behaviour

Knowledge and awareness of gullies by the >500 respondents to the household surveys are summarised in Table 5. The data describes typical attitudes about land degradation and the respondents' views on the extent that gullies have advanced between experimental locations, with and without interventions in place. As far as the authors are aware, this represents the first evidence of this type which explores whether local demonstrations of gully treatment effectiveness have an impact on local farmers knowledge, awareness and behaviour. To a general question about the extent of land degradation and gully erosion, there was no evidence of a change in attitudes before and after the experiments took place. However, in the treatment areas, there was some evidence of changes in perception regarding the extent and most important types of soil erosion and where gully erosion occurs, with a greater perception that main impacts are focussed on farmland.

**Table 5.** Impact of on-farm field experiment - Difference in Difference Estimates

|  | Treated-Non-treated | Standard Error |
|---|---|---|
| Do you think there is a land degradation problem in your community? (1- Yes, 0 – No) | -0.207 | (0.120) |
| Compared to your past experience, how do you see land degradation in general now? (1-Increasing, 2-Same, 3-Decreasing) | 0.013 | (0.069) |
| What is the most important form of land degradation? | | |
|   Soil erosion on farmland (1- Yes, 0 – No) | 0.275 | (0.026)** |
|   Soil erosion on communal grazing land (1- Yes, 0 – No) | -0.108 | (0.127) |
|   Gully erosion (1- Yes, 0 – No) | -0.174 | (0.054)** |
|   Depletion of soil quality [SOM and nutrient depletion] (1- Yes, 0 – No) | 0.002 | (0.076) |
| Compared to your past experience, how do you see gully formation in general now? (1-Increasing, 2-Same, 3-Decreasing) | -0.047 | (0.129) |
| Where are gullies most frequently observed? | | |
|   Farmland (1- Yes, 0 – No) | 0.246 | (0.087)* |
|   Grazing land (1- Yes, 0 – No) | -0.087 | (0.093) |
|   Communal land (1- Yes, 0 – No) | -0.159 | (0.180) |
| Do you think the watershed activities used help control gully formation? (1-Yes, 0-No) | 0.049 | (0.101) |
| What measures are you aware of that are taken to control further gully formation? | | |
|   Tree planting (1- Yes, 0 – No) | -0.101 | (0.055) |
|   Watershed activities (1- Yes, 0 – No) | -0.131 | (0.027)** |
|   Terracing (1- Yes, 0 – No) | 0.117 | (0.068) |

**Note**: The treated-non-treated estimates represent the estimated coefficient on the variable for the kebele from a simple fixed effects difference in difference estimation using the matched 2021 and 2023 sample. The Standard errors allow for clustering at the kebele level. * and ** indicate differences which are statistically significant different at 10% (i.e., 90% of confidence interval) and 5% (95% of confidence interval) respectively. There were 522 respondents in 2021 and 500 respondents in 2023.

Table 6 reports the estimates using questions specifically asked in the 2023 survey and represent the difference between the mean in the treated and non-treated areas. Here the recall questions consider whether respondents were aware and able to recollect different land degradation measures from a general cue (i.e. from the list of land restoration measures presented to the respondents). The differences in these scores between the treated and non-treated areas are reported in the first panel of

Table 6. Some differences in the levels of recognition were apparent, such as greater recognition of mountain measures such as terracing, high recognition but with weak evidence (p-value 0.12) of gully measures, and lower recognition of grassland management measures.

Through these household surveys we confirmed our hypothesis that demonstrating interventions through field studies influenced knowledge and understanding of similar gully treatments among respondents. The average measures of effectiveness were all higher for those living where demonstration gullies were established, especially for the gully shaping and check dams. Testing the set of questions (items) using Cronbach's alpha confirms that they can be interpreted as capturing a single underlying characteristic, i.e., knowledge of the gully treatments (alpha 0.74). Testing the averaging aggregate score

also suggests that there was a statistically significant difference between the knowledge and understanding of these measures in the area where the on-field experiments took place.

The last panel in Table 6 reports the results from the multinomial logit estimation applied to the questions (items) on an individual's capacity to deal with different gully treatments (see supplementary materials 2). The estimates reported represent the marginal effects of being in the treated area on the probability of selecting a particular option. For the gully head treatment,

for example, respondents in the treatment area selecting "could do this alone" had a 0.085 lower probability than for those respondents in the areas where no experiments took place. Overall, there was evidence that the respondents in the treated area have a different view of the resources required to undertake the measures. The probability of that the respondent choose the option "could do it on their own" or "need neighbours" was lower in the treatment areas in all cases except for the reshaping of gully banks treatment, and higher for the option "need community mobilization" with the later underlying difference being

statistically significant (at 5%) in 3 out of the 5 different treatments. Hence, the respondents appeared generally more pessimistic of their ability to undertake this alone or with neighbours. This emphasises that increased knowledge of gully treatments may mean farmers become aware of the challenges required to implement treatments.

The final set of questions in the last panel for Table 6, assessed possible differences in behaviours related to gullies between the sample (i.e. household survey respondents) in the areas with and without the on-farm experiments. Here there are

statistically significant differences consistent with greater effort being made to undertake work to reduce gullies in the treated area. The averages for all questions were statistically significantly higher ($p < 0.05$) for the respondents in the treatment area, e.g., the average was 0.397 point higher for question involving individual work and work with neighbours and friends, and 0.248 higher for the question around farmers actively trying to decrease gully formation. This is consistent with the hypothesis that the on-farm experiments and field day demonstration have positively affected farmers behaviour (or intention to) to act to

help restore and prevent gullies (despite their apparent greater pessimism about their capacity to do so alone).

**Table 6**. Impact of on-farm field experiment. Differences in Recall, Recognition and Capacity to deal with Gullys across Treated and Non-treated areas (2023 Survey).

| | Treated-Non-treated[1] | St. Error |
|---|---|---|
| **Recall Awareness Scores** | | |
| Hillside SWC- terraces etc | 0.023 | (0.008)* |
| Farmland SWC - bunds etc | 0.000 | (0.017) |
| Gully Rehab. measures | 0.024 | (0.011) |
| Afforestation/reforestation | -0.041 | (0.021) |
| Biological measure grass planting | -0.027 | (0.039)* |
| Water harvesting structures | 0.020 | (0.008)** |
| Grazing land management | 0.042 | (0.009) |
| **Recognition Effectiveness of Different Treatments (Likert Scale 1-5) [2]** | | |
| Gully head treatment –using stone rip rap/rubble at gully head | 0.121 | (0.072) |
| Reshaping the gully banks at 45º and planting forage grasses | 0.206 | (0.038)** |
| Making check dams made of relatively small rocks | 0.118 | (0.042)* |
| Making check dams constructed using vegetation or logs | 0.202 | (0.044)** |
| Making small barriers constructed of a series gabion baskets | 0.067 | (0.053) |
| Aggregate – Gully head + Reshaping Gully Banks | 0.164 | (0.055)* |
| Aggregate - overall | 0.134 | (0.032)** |
| **Capacity to deal with different Treatments[3]** | | |
| Gully head treatment –using stone rip rap/rubble at gully head | | |
|     Could do it on own | -0.085 | (0.019)** |
|     Need neighbours | 0.034 | (0.033) |
|     Need community mobilization | 0.050 | (0.035) |
| Reshaping the gully banks at 45º and planting forage grasses | | |
|     Could do it on own | 0.030 | (0.047) |
|     Need neighbours | -0.079 | (0.001)** |
|     Need community mobilization | 0.049 | (0.046) |
| Making check dams made of relatively small rocks | | |
|     Could do it on own | -0.045 | (0.023)** |
|     Need neighbours | -0.113 | (0.039)** |
|     Need community mobilization | 0.158 | (0.053)** |
| Making check dams constructed using vegetation or logs | | |
|     Could do it on own | -0.044 | (0.030) |
|     Need neighbours | -0.151 | (0.012)** |
|     Need community mobilization | 0.195 | (0.042)** |
| Making small barriers constructed of a series gabion baskets | | |
|     Could do it on own | -0.021 | (0.011)* |
|     Need neighbours | -0.116 | (0.015)** |
|     Need community mobilization | 0.137 | (0.011)** |
| **Behaviour (5-point Likert scale) [4]** | | |
| In the last 6 months, I have undertaken work on my own or with neighbours and friends to help restore and prevent gullies on the land which I use | 0.397 | (0.024)** |

| | | |
|---|---|---|
| In my farming, I actively try to decrease gully formation | 0.248 | (0.024)[**] |
| In the last 6 months, I have undertaken work as part of the community to help restore and prevent gullies | 0.282 | (0.013)[**] |

[1] For each variable the reported coefficients represent the average difference in responses between the treated and non-treated. [2] Scale 1-Not at all effective, 2-Not really effective, 3-Somewhat Effective, 4-Effective, 5-Very Effective. [3] These estimates represent the marginal effect of being in the treated area on the probability of selecting the option - *Could do it on own, Need neighbours, Need community mobilization.* The underlying coefficients for this calculation are estimated using a simple multinomial logit with treatment as the only covariate. [4] Scale 1-No, not at all, 2-Yes, once or twice, 3-Yes, occasionally (more than twice), 4- Yes, regularly (at least once a month). 5-Yes, very often (at least once a week). The Standard errors allow for clustering at the kebele level. [*] and [**] indicate differences which are statistically significant different at 10% and 5% respectively. Data based on 500 responses. SWC – soil water conservation.

## 4 Discussion

The zero upward expansion of gully heads in the treated gullies reflects the effectiveness of interventions in halting the upward expansion of gully heads and reducing soil loss. As all investigated gullies were established in sites exhibiting similar site and climatic characteristics, good reproducibility of the impacts was evident for this locality (Figure 1). The effectiveness of gully head rehabilitation measures we found have been observed in other research based in Ethiopia (e.g. Ayele et al. 2016; Asfawesen et al. 2021). These studies indicated that the key role of gully rehabilitation measures in reducing runoff and soil loss was attributed to the capacity of the measures to reduce the speed of runoff, retain soils and facilitate the growth of grasses in the retained soil. The contribution of check-dams to cultivated and grazing land management was mainly through converting degraded areas into productive land or reducing degradation (i.e. increasing access to or availability of productive land) and improving the production of livestock feed and crops. Despite the positive effects of interventions to halt gully erosion, the key informants believe that the efforts were not adequate. This argument could be attributed to the extent and severity of gully erosion and to the limited scaling-out of activities. Clearly, local measures are accessible to farmers providing mitigation takes place early, before gullies grow too large.

However, addressing the key constraints of out scaling gully rehabilitation measures is required to promote a wider adoption of interventions and long-term sustainability. This, in turn, implies the need to focus on context specific technologies that are socially, economically and environmentally feasible. For example, the increasing prices of gabion wires and the limited access to finance suggest that interventions or technologies for gully rehabilitation should use limited external inputs or be designed based on local knowledge and available local materials. The results also suggest that beginning gully reclamation at an early stage is important, as reclaiming big gullies with other measures, such as loose rock and gabion check dams, is costly and difficult for smallholder farmers to manage, considering their financial and technical capacities. In line with this, a study by Addisie et al. (2018) demonstrated that stopping the uphill advancement of gully heads by reclaiming gullies at an early stage of their formation and development (i.e. while they are shallow) is effective, less costly than reclaiming a bigger gully and can easily be managed by local communities. A study by Nicholson et al. (2015) suggested that the planning, design and implementation of gully rehabilitation measures can be improved through meaningful participation of local communities and co-creation of knowledge.

Differences between men and women in the perceived costs and benefits of gully rehabilitation measures (e.g. men attached higher cost for gabion check dams than women) (Figure 6) could be attributed to differences in their responsibilities in managing financial matters. In the study area, men usually took more responsibility in managing financial resources, which could make them more sensitive to the cost of materials. This implies that implementing interventions needs to consider

perceptions of the benefit factors that vary between different groups of a community. This indicates the need for inclusive planning, design and implementation of gully rehabilitation measures to take diverse opinions into account and ensure sustainability. The results also suggest that different groups appreciate different benefits depending on experience and responsibilities. For example, improvement in access to and availability of water is more important for women than men, as women are usually responsible for water collection.

The results also show that generating short-term economic benefits that are easily demonstrated is key to sustaining the assessed gully rehabilitation measures, even though associated tangible non-economic benefits and ecosystem services are appreciated. The negative overall assessment of gabion check-dams by the local communities was mainly attributed to the high cost of input materials and the technical requirements, which were often beyond the capacity of local communities or farmers.

The gully erosion interventions deployed in this study are low-cost and practical for application in any region. Because the

510 tested gully rehabilitation measures mainly use local materials, and require limited technical and financial inputs, they were socially acceptable and could be implemented within the farmers' capacity. By combining understanding on impacts to people with environmental benefits, best practices that are most likely to be deployed and maintained have been identified. Further research in other regions will disentangle the global significance, but we found good agreement between the effectiveness of interventions in the Halaba district, to what has been observed in studies exploring similar interventions in other regions. For

example, a study in the Ethiopian highlands demonstrated that investment in low-cost gully rehabilitation measures can be an economically viable proposition (Yitbarek et al. 2012). It further suggested that stakeholders involved in gully rehabilitation should continue to invest in appropriate techniques of gully rehabilitation and management to ensure continued benefits from rehabilitated gullies and use of surrounding farmlands. In a study in Pakistan, Hussain et al. (2022) also suggested that using gully rehabilitation technologies that are low-cost and easy to implement is key for wider adoption. Communities have a strong

desire to act to improve their farming livelihoods, with an appreciation of environmental benefits too. Effective policy development, therefore, should focus on small-scale, cost-effective interventions, over more costly systems, such as gabion baskets that are beyond the means of local resources.

## 5 Conclusions

The results suggest that the tested low-cost gully rehabilitation measures were effective and are viable options to mitigate the upward expansion of gully heads, thereby significantly reducing soil loss. Farmers also indicated that the implemented gully rehabilitation measures were successful, though out-scaling or wider implementation of these gully rehabilitation measures were constrained by diverse social, economic and environmental challenges. However, this could be addressed by context specific technologies that are feasible to implement. Introducing incentive mechanisms to motivate communities to participate in gully rehabilitation activities is also key to strengthening the protection of gully rehabilitation measures so that the potential benefits are sustained. Furthermore, as the benefit factors of an intervention cannot be the same for different groups of a community, it is crucial to have inclusive planning, design and implementation of gully rehabilitation measures and take diverse opinions into account.

The approach used in this study (i.e. the integration of on-farm field experiments with socio-economic studies and field demonstration) influenced knowledge and understanding of local communities on available low-cost gully rehabilitation measures and their physical, natural and human assets as well as the time required for interventions. Specifically, there was clear evidence of greater effort to undertake work to reduce gullies in areas where mitigation measures had been implemented, suggesting that the on-farm experiments and field day demonstration had empowered farmers to act despite their greater pessimism about their capacity to do so. In addition, farmer participation in our investigation of gully mitigation strategies supported the selection of the most effective, acceptable and practical interventions from the perspective of local communities. Communities have a strong desire to act to improve their farming livelihood, with an appreciation of environmental benefits too. Effective policy development, therefore, should focus on small-scale, cost-effective interventions, as over engineered systems, such as gabion baskets, are beyond local resources.

**Authors contribution**. WM: writing – original draft, data collection, analysis, investigation, visualization, methodology. EP: writing – original draft, data analysis, conceptualization, supervision, validation. GY: data collection, review and editing. DT: data collection, analysis, review. AM: data collection, analysis, review. YT: data collection, analysis, writing – original draft. DM: data collection, analysis, review. CG: data analysis, conceptualization. PDH: writing – review and editing, conceptualization, validation. JS: writing – review and editing, conceptualization, validation.

**Competing interest.** One of the (co-) authors, Prof Paul D. Hallett is a member of the editorial board of SOIL.

**Acknowledgements.** This work was supported by the Economic and Social Research Council grant awarded to the University of Aberdeen (Grant number ES/T003073/1).

**Ethics Application:** The university of Aberdeen submitted ethics application for using human subject and awarded certificate from the university.

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
