# Peer review of "Gully rehabilitation in Southern Ethiopia – value and impacts for farmers."

_EGUsphere, 2024_

## Author Comment (AC2)

**Response to reviewer one**

**General comment:** This manuscript addresses the effect of low-cost gully interventions in farmer communities in Southern Ethiopia. Based on the monitoring of upward expansion of gully heads, the author show that severe gully erosion can be minimized with riprap, grass planting and check-dam integrating. Over a period of 26 month these rehabilitation activities ceased gully head expansion, whiles untreated gully heads expanded 671 cm on average. Informant interviews, group discussions and household surveys show that farmer´s knowledge and perception of their capacity to act against gully erosion increases after on-farm trials and field day demonstrations. Soil degradation due to gully erosion causes persistent problems regarding agricultural and pasture activities especially in Sub-Saharan rural communities. Thus, the topic is relevant to a larger community of readers since it provides insights on how farming communities can be empowered via education and field training to reduce severe soil erosion with low-cost measurements. While the authors are well presenting the relevance of their study, the main issue is that the discussion and presentation of the results are not well-structured and remains too descriptive. I strongly suggest a complete rewrite of the result and discussion section of the manuscript to enhance the main take away message of this study, which is indeed important. The result and discussion section also contains many parts, which belong to the method section. Another flaw of the study is the lack of clear and testable hypothesis to which the discussion section could tight back again. In addition, there are parts in the study design and methodology which needs more clarification. Especially the statistical analysis needs to be described in more detail to comprehend each step in the interview, survey and group discussion evaluation. Even though this study focuses on qualitative analysis, parts of the study design regarding soil physicochemical properties need more clarification. Finally, the conclusion section now is only a short summary of the study but should focus on what the reader have learned after reading this study. Please see my comments on that below. At this stage, the manuscript needs a major revision before considered to be published in EGUsphere.

*Response: We are very grateful for the valuable and constructive comments and suggestions given by the reviewer. We addressed all the comments by providing details, splitting the results and discussion section and fully revising the conclusion. The method section was also restructured, and the biophysical and socio-economic components are discussed separately. All the specific comments are addressed. For more details, please see below the responses given for the specific comments.*

**Specific comments**

**Comment 1**. *Abstract:* In general, the abstract is well structured and presents the main take away messages from the study. There are minor points which need to be addressed.

*Response. We are very grateful for the positive remark. We addressed all comments/suggestions. Please see the responses to the specific comments below.*

**Comment 2**. Line 15: Can you please explain buy-in and input in this context?

*Response. We appreciated this concern. In the context of the study, buy-in and input refers to the alignment of interventions with interest and priorities of local communities and the need to validate the results of biophysical assessments by local communities (please see abstract section, lines 14-15, in the revised manuscript.*

**Comment 3**. Line 18: Do you mean with "in-context changes" the field day training of the farmers?

*Response. The use of the term in the abstract is related to the participation of farmers in on-farm field trials demonstrating gully interventions. Please see line 18, in the revised manuscript.*

.

**Comment 4.** Line 18 and 23: You first mentioned that the measurements are low-cost but then farmers are mentioning the high cost of input material at least for the gabion check-dams. What does low-cost mean in this context? Is the input material cheap or the labor time? And for whom is it low-cost (for the individual farmer, the community)?

*Response. We are grateful for this comment. We revised the sentence and removed descriptions on gabion check-dams as this is not evaluated and not considered as low-cost gully rehabilitation (please see abstract section, lines 23, in the revised manuscript). In this study, low-cost gully rehabilitation measures refer to interventions that can be implemented with farmers capacity or with limited external support (please see page 6, study design section, lines 144-145, in the revised manuscript.*

**Introduction**

**General comment**: The introduction is well-structured and written. It presents the significance of the land degradation problem caused by gully erosion in general and shows why this issue is even more severe in the study region. The author discusses soil erosion measurements and presents the problem, that farmers may perceive such actions as requiring too much labor and high cost to implement them on their own land. While the authors are well presenting the problem and how to potentially mitigate this issue with on-farm experiments combined with qualitative analysis (interviews, group discussions, household surveys), the introduction lacks clear hypothesis based on theoretical reasoning. Please see my further comments below:

*Response: We are very grateful for the constructive feedback. We also addressed all comments (please see the responses given below for the specific comments. We also described the key hypothesis of the study, the objectives and what we wanted to be investigated and achieve. Please see, page 4, lines 104 - 114, in the revised manuscript.*

**Specific comments**

**Comment 1**. Line 30: Give examples for key environmental services.

*Response. Agreed and provided. Also, the phrase "environmental services" is revised as "ecosystem services". Please see page 2, lines 30-31, in the revised manuscript.*

**Comment 2.** Line 33: Give examples for those environmental and anthropogenic factors.

*Response. Agreed and provided. Please see page 2, lines 34-37, in the revised manuscript.*

**Comment 3**. Line 36: Please provide more reference for this statement (global problem of gully erosion).

*Response. Agreed and provided. Please see page 2, line 38, in the revised manuscript.*

**Comment 4**. Line 37: Provide more insights why Africa is the worst-hit continent regarding gully erosion and what the drivers are.

*Response. Agreed and provided. Please see page 2, lines 39-40, in the revised manuscript.*

**Comment 5**. Line 39 – 41: More references needed for this statement.

*Response*: *Agreed and provided. Please see page 2, lines 44-47, in the revised manuscript.*

**Comment 6**. Line 53 – 55: Explain more why the problem is bigger in poorer countries instead of repeating that Ethiopia is challenging gully erosion.

*Response:* *Agreed and addressed. Please see page 3, lines 62-64, in the revised manuscript.*

**Comment 7.** Line 79 – 80: Elaborate this sentence a bit more. What do you mean with biophysical evidence in this context?

*Response*: *Agreed and addressed. Please see page 4, lines 96-97, in the revised manuscript.*

**Comment 8**. Line 86: Hard to follow. Gullies as a simple indicator for gully erosion? What do you mean with simple indicators in this context?

*Response*: *We appreciated this concern and edited. Please see page 4, line 103-104, in the revised manuscript.*

**Comment 9**. Line 88: Give reasoning why your study region and research has a wider applicability to other regions. You need first present your results to make this statement. This statement fits better to the conclusion section.

*Response*: *We agreed with this suggestion and moved the sentence to the conclusion section.*

*Methods*

**General comment**: In general, the method section needs a more detailed description of the soil characteristics and management in the study area and how this affect gully erosion. More information is needed how the land cover, climate gradient and strong rainfall events were treated in the analysis since those factors will most likely impact gully erosion. Also, what initializes gully erosion in the study area (bare soils without any vegetation cover, tillage on steep slopes, cattle)?  See my comments below:

*Response:* *We are very grateful for the constructive comments and addressed. We clarified how land cover, climate gradient and other external factors were managed in our study design. For more details, please see below the responses given for the specific comments.*

**Comment 1**. Figure 1: Where are the study sites located at panel c? The panel a, b and c need more description since it is not clear from the figure alone what they are show.  What about parent material mineralogy and texture? Does differences in soil parent material affect gully erosion?

*Response*: *Figure 1 was revised as suggested and the location of the 9 paired gullies are indicated in the map. We also added an explanation on the figure caption and described what a, b and c are represented. The 9 paired gullies located in similar landscape position with similar features and the areas where the gullies are located do not show considerable differences in soil properties, parent materials and LULC. The 9 paired gullies are distributed within less than 2 hectares of land (Figure 1) and exhibit the same climatic conditions in terms of rainfall and temperature.*

**Comment 2.** Table 1: I would present the last four parameters in a separate table since it shows social-economic information. I also would suggest using the unit ha for presenting the landholding area since it is more common for readers. How is the climate gradient treated in the analysis? More information

needed about the land cover, land use and management and how this impacts gully erosion in the study area.

*Response: We appreciate the reviewer's suggestion and addressed this comment by revising the heading of the first column as "Site and household characteristics", please see Table 1, in the revised manuscript. All the nine paired gullies were selected from mid slope position (Figure 1) with similar characteristics in terms of soil, topography and land use and land cover. Also, the nine paired gullies were spatially distributed within less than two hectares of land area, and do not show any differences in climatic variables, such as rainfall and temperature (this information is included on page 6, lines 135-138, in the revised manuscript).*

**Comment 3**. Line 123: What is the reason for only choosing gullies with depths less than 3 m in the analysis?

*Response: This was done to control the influence of gully size on the effectiveness of the tested low-cost gully rehabilitation measures (please see page 6, line 135-137, in the revised manuscript). In addition, the overall goal of the study was to demonstrate gullies can be rehabilitated by farmers capacities if farmers act at the right time and aware about the available low-cost measures.*

**Comment 4**. Line 131: Better references are needed to describe the soils and their erosion history in the study area since this is important information for the reader and the overall study.

*Response: We appreciated the reviewer's concern. but we presented the primary data in Table 3.*

**Comment 5**. Table 2: Is this table showing the results after 26 months? This would rather belong to the result section.

*Response: The gully dimensions in Table 2 are measured values prior to the establishment of the on-farm trials. This is clarified in the table caption in the revised manuscript.*

**Comment 6.** Table 3: Are you sure this is a Luvisol? Based on the interlayering of horizons with different clay content, it could also be a Fluvisol. I would also suggest using a soil classification considering tropical soils. What is a kebele?

*Response: The soil classification is rechecked and we prefer to maintain as Luvisol. Kebeles are typically localities containing around 500 households, with an administrative centre around which agricultural extension is typically organized. Please see page 8, lines 169-170, in the revised manuscript.*

**Comment 7**. Line 138: What is woreda?

*Response: woreda is equivalent to district and changed in the revised manuscript.*

**Comment 8.** Figure 2: You need to describe treatments in more detail so that the reader can reproduce the method. How do you define a knowledgeable farmer?

*Response: We appreciate the reviewer's concern. But the treatments are described in study design section, pages 6, lines 139-143, in the originally submitted manuscript. Knowledgeable farmers in this study refers to those farmers with experience in organizing and leading watershed management practices, members of community watershed team and early adopters or production cadres (please see page 8, lines 182-184, in the revised manuscript).*

**Comment 9**. Line 167: Where can I see the questionnaire used in this study?

*Response: We provided this as a supplementary materials. Please see supplementary material 1 in the revised manuscript.*

**Comment 10**. Line 171: Did you also monitor the gully heads during and after strong rainfall events? And how was changing vegetation cover considered due to crop growth, harvesting etc.?

*Response: All investigated gullies located in similar locations and do not exhibit difference in site and climatic characteristics. These aspects of the research were controlled using a proper research design.*

**Comment 11**. Line 172 – 173: How exactly was the permanent reference point established? This information is very important. How did you ensure that the reference point was not changed during the monitoring?

*Response: We appreciate this concern. All our reference points were big trees on the farm and are close to the investigated gullies. These big trees are protected and are not allowed to cut down (please see page 6, line 148-149, in the revised manuscript).*

**Comment 12**. Line 176: Why was erosion further down the gully not considered in the study?

*Response: The focus of the study was to investigate the effectiveness of the treatments to controlling the upward expansion of gully heads. Just it was not within the scope of the study.*

**Comment 13**. Foot note 1: Please include the information from the footnote in the text body since it distracts the reader.

*Response: Agreed and revised.*

**Comment 14**. Line 185 – 186: How exactly does the coding scheme work?

*Response: The coding was done manually. Please see page 9, line 198, in the revised manuscript.*

**Comment 15.** Line 183: What is deductive coding?

*Response. We addressed this by incorporating explanation. Please see page 9, lines 198-200, in the revised manuscript.*

**Comment 16**. Lines 195  198: Please explain this approach in more detail.

*Response: We agreed with this comment and discussed (please see pages 9-10, lines 215-223, in the revised manuscript. We also provided supplementary material 2 for more details.*

**Results and discussion**

**General comment**: This section is the main flaw of this manuscript. It is poorly structured and contains many parts, which belong to the method section. This section needs to be divided in an individual result and discussion section.  In general, this section is too descriptive and does not discuss the results in a bigger context. More references are needed. The discussion should then focus what the reader can learn from this study.  In addition, the author needs to show that their field measurements are the dominant

factor influencing gully erosion since the study area also has a gradient in climate and soil properties and most probably changes in crop cover over the seasons. The comments below:

*Response: We are very grateful for the constructive feedback. In the revised manuscript, we presented the results and discussion in two separate sections and discussed the bigger picture of the study. We also clearly showed that the design of the study is able to control potential external factors affecting the results. For more details, please see the responses given to the specific comments.*

**Comment 1:** Line 206: Please present the also the standard deviation from the mean value.

*Response: Agreed and provided, please see page 11, lines 256, in the revised manuscript.*

**Comment 2:** Line 206 – 207: This needs to be mentioned in the method section first. Please also report standard deviation. How exactly was soil erosion calculated?

*Response: This has been discussed in the originally submitted manuscript. However, we elaborated further in the revised manuscript. Please see page 6, lines 150-153, in the revised manuscript and the standard error was also provided (page 11, line 257).*

**Comment 3**: Line 207 – 209: What about other factors which could impact gully erosion besides the interventions (like climate, land use management, crop cover, differences in soil properties etc.)? You need to show that the effect of halting upward expansion of gully heads is mainly driven by the treatments.

*Response: We very much appreciate this concern. We addressed this during the design of the research. We established treatments in similar sites and the investigated gullies were not very much spatially distributed. We elaborated this in the revised manuscript. For example, please see page 6, lines 135-138.*

**Comment 4:** Figure 3: How do you explain the sudden increase of gully erosion after 26 months in all control plots? How do you explain the differences in gully erosion with time across control plots? What do you mean with "(…) the gullies were spatially distributed, and a single control cannot work"? Are they different in soil properties, rainfall intensity, amount of cattle etc., which could affect gully erosion?

*Response: The last measurement was taken after 26 months and after nine months of taking the previous measurement. Within this period, the gullies experienced two rainy seasons (the short and long rainy season). The measurements show the cumulative effect and with time, the control gullies are exposed to several environmental conditions and loosen with time. This might cause a sudden increase in upward expansion. The sites are not different in terms of major soil types, land use and land cover and environmental condition but it is always better to have the closest gully as a control than using a single control gully for all treatments.*

**Comment 5**: Line 216 – 218: Do you mean that the amount and the depth of gullies are different across slope positions?

*Response: This refers difference in the number and density of gullies; density refers to the number of gullies in an area.*

**Comment 6:** Figure 4: What is the meaning of the yellow arrows? What are SWC measures? What are the differences between drivers and pressures?

*Response:* We appreciate this concern. The yellow arrows in figure 4 show that a response, which is the result of an undesired impact can affect any part of the chain between driving forces and impacts. This is described in the revised manuscript (please see the caption of figure 4 in the revised manuscript). The SWC refers to soil and water conservation and is written in full in the revised manuscript. Drivers are usually the need for individuals, industry or macroeconomic strategies as well as the inherent properties of a site. Driving forces lead to human activities that result in meeting a need. This, in turn, exert pressure on the environment. Generally, pressures can be categorized as excessive use of resources, land use and land cover changes and emissions.

**Comment 7**: Line 252 – 253: This sentence is hard to follow. What do you mean by that statement?

*Response:* Revised. Please see page 13, line 301-302, in the revised manuscript.

**Comment 8:** Line 254: How can land affected by gully erosion converted into productive land? The measurements used in this study only stop the upward extension of already existing gully systems.

*Response:* We appreciate the concern. However, the study has two components, biophysical measurement (upward expansion of gully heads) and qualitative assessment of the functioning of the system and benefits and disbenefits of interventions. This section discusses the benefits and disbenefits of gully rehabilitation measures from the perspective of local communities. After the rehabilitation of gullies, affected land can be used for productive purposes. For example, growing forage tree and grasses, fruit trees, etc. This has been discussed in detail in sections 3.2 and 3.3.

**Comment 9**: Line 256 – 258: This is a circular argument which needs revision.

*Response*: Revised. Please see page 13, line 302-305, in the revised manuscript.

**Comment 10**: Line 259 – 260: Again, how can land already affected by gully erosion converted into productive land? How is accessibility related to soil degradation in this context?

*Response*: Please see the response for comment 8 regarding the conversion of bad lands to productive lands through the implementation of gully rehabilitation measures. Land degradation constrain the access to productive lands or reduce availability of productive lands.

**Comment 11**: Line 263: If there is sediment accumulation happening, after implementing the rehabilitation measures, then there is still soil erosion happening. Thus, this argument needs revision. And how does this go together with your results, showing no gully erosion anymore after the implementation of the measurements?

*Response*: We appreciate this concern. However, in the agricultural landscape of the watershed, yes there is still rill and sheet erosion, and sediments can be accumulated at the back of check dams. The previous discussion was on the effectiveness of interventions to reducing/halting upward expansion of gully heads. This discussion does not contradict the previous one.

**Comment 12**: Line 270: How are gullies restored into productive land?

*Response:* Gullies can be converted into productive land. What is key here is identifying when to act and what technologies we have. This research also demonstrated how low cost-gully rehabilitation measures are effective in halting gully expansion. Specifically, the reshaping of gull banks and planting grasses and forage trees on the banks support to convert those affected lands to productive land.

**Comment 13**: Figure 5: What does "Promoted changes in cultivated and grazing land management" mean?

*Response: This refers to improvement in cultivated and grazing land management. We revised it as "improved cultivated and grazing land management". Please see figure 5, in the revised manuscript.*

**Comment 14:** Line 277 – 278: Does this mean that the farmers are still dependent on external help after the implementation? How does this come together with your argument that farmers capacity to act by their own after attending to your field demonstrations?

*Response. This again does not contradict. This statement is about enhancing effectiveness. Always, updating the local communities is key though they have basic skills.*

**Comment 15**: Line 279 – 280: This is an important statement from the respondents, even though it is a small fraction of 10 %. You should elaborate on why this is the case.

*Response: Agreed and elaborated. Please see the discussion section, page 21, lines 481-484, in the revised manuscript.*

**Comment 16**: Line 301 – 302: Even though this is an important point, how does this relate to gully rehabilitation in detail? What do you mean with incentive mechanisms?

*Response: Agreed and elaborated. Please see page 14, lines 349-352., in the revised manuscript.*

**Comment 17**: Line 310 – 313: This is important information and need to be discussed in more detail. What can we learn from this? Based on this outcome, how could the overall gully rehabilitation measures be improved?

*Response: Agreed and added. Please see the last paragraph of the discussion section.*

**Comment 18:** Line 318 –322: What are the drivers behind? Have such trends been observed in similar studies before?

*Response: Addressed by adding some explanation. Please see discussion section, page 21, lines 485-489, in the revised manuscript.*

**Comment 19**: Line 326 – 334: What are the reasons and what can we learn from that. This is too descriptive and need to be discussed in more detail.

*Response: Addressed by discussing the driving forces and implications for future actions. Please see discussion section, page 21, lines 485-489, in the revised manuscript.*

**Comment 20.** Figure 6 and Figure 7: What is the difference between those figures? What are the takeaway messages? Overall, these figures need to more explanation in the caption.

*Response: Figure 6 presents the aggregate values of benefits and cost of all the three categories (environmental, economic, and social). Figure 7 present disaggregated values of benefits and cost. This explanation is added in the figure caption, please see figure 6 in the revised manuscript.*

**Comment 21:** Line 369 – 371: This is not a strong argument, but you mention this in the abstract as a main message of this study (empowering farmers to act).

*Response: We appreciate this concern. in the revised manuscript, this is elaborated in the last paragraph of the discussion section and in the conclusion.*

**Comment 22**: Table 5: What does "Treated-Non-Treated" mean? In general, this table needs more explanation. Do positive values mean yes-answers and negative numbers no-answers in the interview? What is the reason for a p-value <0.1? What is the difference to table 6?

*Response: The treated – Non-treated, in the study, refers to experimental locations with and without interventions. This is also indicated in data collection and analysis section of originally submitted manuscript.*

**Comment 23**: Line 380 – 382: This belongs to the method section; Line 383 – 386: This belongs to the method section; Line 390 – 396: This belongs to the method section; Line 397 – 403: Again, this belongs to the method section. You need to explain your statistical analysis step by step first. Otherwise, it is hard for the reader to comprehend your results and interpretations. What do you mean with "a single underlying characteristic"? What is the alpha-value? What is an averaged aggregate store? Line 404 – 409: See comment above. This belongs to the method section. How does a simple multinomial logit model work?

*Response: We agreed with this comment and moved texts to the method section.*

**Comment 24:** Line 383: Please specify what you mean with a "general cue" in this context.

*Response: This refers to from the list of land restoration measures presented to the respondents. Addressed in the revised manuscript.*

**Comment 25**: Footnote 2: This belongs to the method section. No footnotes please since it interrupts the reading flow.

*Response: This is deleted, and the text is included in the method section.*

**Comment 26**: Line 409 – 410: How does the evidence look like? What can we learn from that? What are similar studies say about that?

**Response**: We have clarified the nature of evidence expanding the discussion of these results lines 442-456, and emphasized, the novelty of these results at the beginning of this section.

**Comment 27**: Line 412: To which sample do you refer to?

*Response: This refers to household survey respondents.*

Comment 28: Line 414: How does this clear evidence look like? In this information shown in table 6?

*Response: As above, we have clarified the nature of evidence expanding the discussion of these results lines 442-456.*

**Comment 30**: Line 416: This is not clear. Are the farmers feel empowered after the field day demonstration or are they still pessimistic? And how does this fit to the above-mentioned clear evidence?

*Response*: *As above, we have clarified the nature of evidence expanding the discussion of these results lines 442-456. We also mention that the evidence.*

**Comment 31**: Line 418 – 424: Explain why your applications are practical for any region. Based on what data is your application most likely to be deployed and maintained? You mentioned before that some farmers are still pessimistic and depend on external help to implement the erosion measurements. You conclude that your study is in good agreement with similar studies. However, this needs to be discussed in more detail using more references than just one sentence at the end of the section.

*Response:* *We expanded this and provided further discussion. Please see page 22, lines 510-515, in the revised manuscript.*

**Conclusion**

**Comment 1**: The conclusion needs to be rewritten. Now the conclusion is just a short summary of the study.

*Response*: *We agreed and revised the conclusion. Please see the revised conclusion on page 23.*

**Competing interests**

**Comment 1**: Which authors exactly are members of the editorial board? Be transparent with this.

*Response*: *Agree and addressed.*

---

## Author Comment (AC3)

**Response to Reviewer 2**

**General comment**: Thank you for your submission. This was an interesting manuscript that fits within the scope of SOIL. Greater attention to detail in certain areas of the manuscript will greatly improve its readability. It seems like a lot of generalities are stated in the intro, "global problem", "worsening effects" but not a lot of details is provided to back up these claims. Some additional measures of the environmental and economic impacts of gully erosion would really be helpful. The results and discussion section was difficult to follow, I tend to find combined sections to be this way. The results of the gully remediation methods don't seem to be the real takeaway but the impacts and perceptions of local communities? Could you restructure the results/discussion to make these points more clearly? Finally, since you are working with human subjects, did go through institutional review boards? I think that some statement to this effect is needed.

*Response: We are very grateful for the constructive feedback. We addressed all the comments and suggestions and revised our manuscript. We also presented the results and discussion section in separate sections. For more details, please see below the responses given for the specific comments.*

**Specific comments**

**Introduction**

**Comment 1**: Line 36: Could you provide some more context for gully erosion as a global problem? Contributions to marine/water pollution? Land degradation?

*Response: We appreciated this concern. but the global context is discussed in the first paragraph. Please see page 2, lines 29-37, in the revised manuscript. The second paragraph focuses on the problem in SSA.*

**Comment 2**: Lines 38-40: are these on the ground measurements or from model estimates? Could you specify?

*Response: The values are based on ground measurements, and we clarified this in the revised manuscript (please see page 2, lines 41-44 in the revised manuscript).*

**Comment 3**: Line 40: What are the economic costs? Could you provide an actual value to drive home your point about the high cost?

*Response: We agree with this comment and provided some examples of actual values. Please see page 2, lines 45-47, in the revised manuscript.*

**Comment 4**: Line 48: change to "Across different countries" and delete "of the world". And what initiatives and where?

*Response: We removed this section in the revised manuscript for a better flow of information and to shorten the introduction section.*

**Comment 5**: Line 50: change big to large. And how much money and labor are required? It would provide context for the scale of the problem.

*Response: We agreed with this comment and addressed.*

**Comment 6**: Lines 53-55: These two sentences seem out of place and almost read like they are the start of another paragraph/thought. Should they be added to the paragraph below?

*Response: We revised the sentences, please see page 3, lines 62-64, in the revised manuscript.*

**Comment 7**: Lines 61-62: How much worse? Again, are there measures or statistics from these references to bolster your sentence?

*Response: We agreed with this comment and addressed. Please see page 3, lines 72-76, in the revised manuscript.*

**Comment 8**: Line 63: What do you mean by context specific? Could you clarify?

*Response: This refers to measures that consider availability of resources and capacities of local communities, among others. This is included in the revised manuscript. Please see page 3, line 76-77.*

**Comment 8**: Line 80: Add "on" after the word "demonstrating"

*Response: Added.*

**Comment 9**: Line 86: change "big" to "large"

*Response: Changed.*

**Methods**

**Comment 1**: Table 1. Demographic information would seem more appropriate to present in a separate table from the geophysical parameters.

*Response: We appreciated this concern and revised the heading of the first column as site and household characteristics. This helps to reduce the number of tables. Please see page 6, Table 1 in the revised manuscript.*

**Comment 2:** Table 2. Can you change z to "elevation" or "altitude" and add units. Can you also add units to x and y values? Are these lat or long?

*Response: We agreed with this comment and addressed. Please see Table 2 in the revised manuscript.*

**Comment 3**: Line 132-134: I think you mean higher clay content. Also, it is interesting that this soil has alternating Bt and E horizons? I'm assuming this is a loess-derived soil and that these are likely different loess packets?

*Response: This could be possible as there is deposition in the mid slope positions where the experiments are established.*

**Comment 4**: Line 140: you mean "comprising".

*Response: Yes and corrected.*

**Comment 5:** Lines 151-169: I had a very difficult time understanding who was surveyed, where, and why. Especially once I started reading the results and discussion section. You had focus groups and a community-wide survey? In the community-wide survey? Some of these were in kebele with rehabilitated gullies and others weren't? Or were they all around gully rehabilitation projects and you were surveying them pre- and post-intervention?

*Response: We appreciate this concern. To make this clear, we reorganized the method section and provided sub-headings for the different sections. The household surveys were done in the treated and untreated kebeles. This is discussed in detail and clarified in the revised manuscript. Please see section method section in the revised manuscript.*

**Comment 6**: Line 162: change "big" to "large".

*Response: Changed.*

**Comment 7**: Line 161-169: the description of the respondents is not clear as written.

*Response: We appreciate this concern and revised. Please see section 2.3 in the revised manuscript.*

**Comment 8**: Line 178: Can you not repeat key twice in one sentence? I know the discussion points and informants are important but you have already established this.

*Response: Addressed.*

**Results and discussion**

**Comment 1:** Line 206: Why assume a bulk density? You could use a pedotransfer function from your soil data to estimate a bulk density. See Rawls 1983 or similar papers for this.

*Response: We just assumed a typical range of bulk density of agricultural soils, as the aim was to provide estimated figures on soil loss.*

**Comment 2**: Line 301: What do you mean by free riders?

*Response: In this context, it refers to persons who want to benefit more by violating existing rules and regulations. This is clarified in the revised manuscript (page 14, line 349)*

**Comment 3**: Line 337-338: I'm not sure I follow this sentence? So benefits were mainly environmental and costs were mainly social?

*Response: The results suggest that benefit factors are mainly categorized as environmental and economic, whereas cost factors are largely related to social and economic costs. This is addressed in the revised manuscript. Please see page 16, lines 388-390, in the revised manuscript.*

**Comment 4**: Figure 6: What does the number refer to? Is this the sum of the ratings? I also think that since the mean rating can only between 1-3, you should have these on separate figures. Or a two part figure.

*Response: We added more explanations in the figure caption in the revised manuscript.*

**Comment 5:** Table 5. Should you say treated-non-treated or pre and post intervention? Or are you comparing responses to areas from which you did not before any gully rehabilitation to areas that you did? Also how did you test significance?

*Response: Yes, the surveys were done in treated and non-treated kebeles. These issues are addressed in the revised manuscript. Please see the study design and data collection and analysis sections.*

**Comment 6**: Section 3.4 The description of the survey questions presented in Table 6 read more like methods and should be moved there. Also was this the same group of people that were surveyed in the community wide survey that results were presented in Table 5?

*Response: Agreed and revised. We moved texts describing the methods and approaches to the data collection and analysis section in the revised manuscript. We also expanded the discussion and provided supplementary materials.*

**Comment 7**: Lines 418-424: This paragraph seemed likes a better conclusion than the current concluding paragraph.

*Response: We appreciate this suggestion. We revised the conclusion section in the revised manuscript. Please see page 23 in the revised manuscript.*

**Comment 8**: Acknowledgements. Who was the grant awarded to? Don't you also need to include some statements about IRB for using human subjects?

*Response: This is addressed by adding the required information and ethics application. Please see page 24, acknowledgement and ethics application.*